

# Success of concrete and crab traps in facilitating Eastern oyster recruitment and reef development

Emma E. Johnson[1], Miles D. Medina[2], Ada C. Bersoza Hernandez[1], Gregory A. Kusel[1], Audrey N. Batzer[3] and Christine Angelini[1]

[1] Department of Environmental Engineering Sciences, Engineering School of Sustainable Infrastructure and Environment, University of Florida, Gainesville, FL, USA
[2] Department of Agricultural and Biological Engineering, University of Florida, Gainesville, FL, USA
[3] School of Natural Resources and the Environment, University of Florida, Gainesville, FL, USA

Corresponding author
Christine Angelini,
c.angelini@ufl.edu

## ABSTRACT

**Background:** Abundance of the commercially and ecologically important Eastern oyster, *Crassostrea virginica*, has declined across the US Eastern and Gulf coasts in recent decades, spurring substantial efforts to restore oyster reefs. These efforts are widely constrained by the availability, cost, and suitability of substrates to support oyster settlement and reef establishment. In particular, oyster shell is often the preferred substrate but is relatively scarce and increasingly expensive. Thus, there is a need for alternative oyster restoration materials that are cost-effective, abundant, and durable.

**Methods:** We tested the viability of two low-cost substrates—concrete and recycled blue crab (*Callinectes sapidus*) traps—in facilitating oyster recovery in a replicated 22-month field experiment at historically productive but now degraded intertidal oyster grounds on northwestern Florida's Nature Coast. Throughout the trial, we monitored areal oyster cover on each substrate; at the end of the trial, we measured the densities of oysters by size class (spat, juvenile, and market-size) and the biomass and volume of each reef.

**Results:** Oysters colonized the concrete structures more quickly than the crab traps, as evidenced by significantly higher oyster cover during the first year of the experiment. By the end of the experiment, the concrete structures hosted higher densities of spat and juveniles, while the density of market-size oysters was relatively low and similar between treatments. The open structure of the crab traps led to the development of larger-volume reefs, while oyster biomass per unit area was similar between treatments. In addition, substrates positioned at lower elevations (relative to mean sea level) supported higher oyster abundance, size, and biomass than those less frequently inundated at higher elevations.

**Discussion:** Together, these findings indicate that both concrete and crab traps are viable substrates for oyster reef restoration, especially when placed at lower intertidal elevations conducive to oyster settlement and reef development.

## INTRODUCTION

The Eastern oyster, *Crassostrea virginica*, (hereafter "oyster") is a commercially important species that has been harvested for several centuries (*Kirby, 2004*; *Lotze et al., 2006*). High, sustained demand for oysters has resulted in overexploitation of their reefs (*Rothschild et al., 1994*; *Jackson, 2001*; *Jordan & Coakley, 2004*), pressure that has acted with other stressors—climate change (*Wright et al., 2005*; *Easter Oyster Biological Review Team (EOBRT), 2007*; *Talmage & Gobler, 2009*; *Levinton et al., 2011*), disease (*Kennedy & Breisch, 1981*; *Berrigan, 1990*; *Jordan & Coakley, 2004*; *Carranza, Defeo & Beck, 2008*; *Powell et al., 2012*), and coastal development and pollution (*Mearns et al., 2010*; *Jackson, 2001*; *Easter Oyster Biological Review Team (EOBRT), 2007*; *Bilkovic & Roggero, 2008*)—to precipitously reduce their abundance. As a result, an estimated 85% of oyster reefs worldwide have been lost in the last 130 years (*Beck et al., 2011*). The dramatic and widespread loss of oyster reefs is concerning not only because of their importance as commercial fisheries (*Peterson, Grabowski & Powers, 2003*; *Rindone & Eggleston, 2011*; *Pierson & Eggleston, 2014*) but also because of the estimated \$550,000–\$9,900,000 $km^{-2}$ $year^{-1}$ they provide in ecosystem services (*Grabowski et al., 2012*)—including habitat for commercially and recreationally important fish and invertebrates, carbon sequestration, shoreline stabilization, and improvement of water quality through filter-feeding (*Grabowski & Peterson, 2007*).

To counteract oyster reef decline, over 1,700 oyster restoration projects covering more than 51.99 $km^2$ of coastal habitat have been implemented across the US Atlantic and Gulf Coasts alone in the past half-century (*Bersoza Hernández et al., 2018*). In areas where oyster larval delivery is not a limiting factor, reef establishment is often limited by the availability of stable substrate. Oyster-based substrates—including oyster shell bags, loose shell, spat-on-shell, and combinations of other materials with oyster shell—are the most common, accounting for 84% of the reef area constructed along the US Atlantic and Gulf Coasts (*Bersoza Hernández et al., 2018*). This preference for oyster-based substrates stems from studies showing that oyster larvae preferentially settle on oyster shell in response to chemical cues emitted from conspecifics (*Bonar et al., 1990*; *Tamburri, Zimmer-Faust & Tamplin, 1992*; *Turner et al., 1994*). However, reliance on oyster shell is problematic because it is becoming increasingly scarce and expensive (*O'Beirn et al., 2000*; *Yozzo, Wilber & Will, 2004*). Furthermore, bioerosion due to boring sponges and ocean acidification, as well as wave action, limit the life span and stability of oyster shell in estuarine environments (*Powell, Kraeuter & Ashton-Alcox, 2006*).

Given the impediments to using oyster shell, restoration practitioners and researchers have been testing other substrates including concrete riprap, limestone, and granite (*O'Beirn et al., 2000*; *Nestlerode, Luckenbach & O'Beirn, 2007*). *Dunn, Eggleston & Lindquist (2014)* evaluated the performance of carbonate-based substrates (oyster shell and limestone marl) and non-carbonate-based substrates (concrete and granite) in North Carolina and recorded similar oyster sizes and growth rates one year after reef construction, suggesting that non-carbonate-based materials hold promise for restoring subtidal reefs, especially in more saline areas where carbonate materials are vulnerable

to boring sponges. In the Gulf of Mexico, surveys of replicate rock-based and shell-based restored reefs revealed that neither rock- nor shell-based restored reefs sustained oyster densities comparable to those observed on natural reefs; but, of these two restoration materials, rock supported greater oyster densities than shell across the region (*La Peyre et al., 2014*). Similarly, *Theuerkauf, Burke & Lipcius (2015)* compared unconsolidated oyster shell, oyster shell embedded in concrete, and interlocking concrete oyster castles, and observed the highest live oyster recruitment, biomass, and densities on the oyster castles. The oyster castles' superior performance was attributed to their higher vertical relief, which prevented siltation and burial in the dynamic intertidal environment *Theuerkauf, Burke & Lipcius (2015)*. Given the variable success of alternative substrates, there remains a need to identify additional cost-effective, easy-to-deploy materials and to quantify their performance in various environmental settings to inform future oyster restoration efforts (*Bersoza Hernández et al., 2018*).

In this study, we explore the potential of two relatively cost-effective, abundant, and durable materials—concrete and wire crab traps—as restoration substrates for the Eastern oyster (*Crassostrea virginica*). Concrete can be recycled from demolished infrastructure or freshly cast in forms that vary in size and roughness, and it has been used in oyster restoration for several decades (*La Peyre et al., 2014*; *George et al., 2015*). While the deployment of concrete as a restoration substrate carries the risk of heavy metal leaching (*Hillier et al., 1999*; *Hartwich & Vollpracht, 2017*), metal content can be substantially reduced through pre-deployment chemical elution (*Höll, 1994*). Wire traps designed for commercial and recreational blue crab (*Callinectes sapidus*) fishing (hereafter, crab traps) are regularly collected by state agencies via derelict trap removal programs. Abandoned traps are relatively abundant, durable, easy to deploy, and cost-effective given their light weight and ability to be quickly stabilized with rebar stakes (*Kreutzer, 2014*). Crab traps—which are sometimes dipped in concrete to enhance their surface area, roughness, and durability—have been utilized for intertidal oyster reef restoration extensively and successfully in South Carolina (*Kingsley-Smith et al., 2012*; *Kreutzer, 2014*). To our knowledge, however, they have never been tested in the Gulf of Mexico where the tidal range is narrower, waters are generally warmer, and oyster exposure to physical and biotic stressors—such as hurricanes and predatory gastropods—differ (*Kimbro et al., 2017*; *Seavey et al., 2011*). Both concrete and crab trap substrates may offer viable alternative solutions for oyster restoration, but their relative efficacy in supporting reef development has never been directly compared. These two substrates differ considerably in factors known to modulate oyster settlement, growth, and vulnerability to predators, including surface area for settlement, chemical cues emitted from their surfaces, shaded surface areas, and effects on water flow (*Michener & Kenny, 1991*; *Turner et al., 1994*; *Bartol, Mann & Luckenbach, 1999*; *Soniat, Finelli & Ruiz, 2004*; *Kuykendall et al., 2015*).

To evaluate the relative performance of concrete and crab traps in oyster reef restoration, we conducted a 22-month field experiment on degraded oyster bars in northwestern Florida. We deployed five replicates of each structure type over a range of intertidal elevations to compare the effects on temporal trends in oyster abundance, final densities of oysters by size class (spat, juvenile, and market-size), and final oyster

reef volume and biomass. We also monitored barnacle densities due to the potential for ecological interactions between barnacles and oysters: Through shell creation, lateral growth, and local effects on biofilm composition and formation, barnacles may potentially facilitate or hinder oyster settlement (*Osman, Whitlatch & Zajac, 1989*; *Boudreaux, 2005*).

We hypothesized that (1) oyster settlement would be higher on concrete structures than crab traps, due to the larger surface area provided; (2) final reef volume and biomass would be higher on crab traps at the experiment's conclusion, potentially due to the higher water flow through the structures and, hence, food delivery to oysters; (3) oyster and barnacle densities would be negatively correlated due to interspecific competition for space; and (4) both crab trap and concrete structures placed lower in the intertidal would support higher oyster recruitment and reef growth than those at higher elevations, due to shorter periods of inundation and food availability.

## METHODS

### Site location and experimental design

We deployed the field experiment in August 2015 on five offshore intertidal oyster bars located at Corrigan's Reef in Cedar Key, Florida (29°09′50″N, 82°59′30″W) for a 22-month period ending in June 2017. These oyster bars, which were all located within a 400 × 200 m area and spaced approximately 30–80 m apart, experience semidiurnal tides with a tidal range of −0.87 to 0.83 m above mean sea level (NOAA predicted tides for Cedar Key, FL: https://tidesandcurrents.noaa.gov). In this region, live oysters are observed at elevations between −0.2 and 0.3 m above mean sea level, and our replicate structures were deployed between 0.1 and 0.25 m above mean sea level. The climate is subtropical with average air temperatures ranging from 13 to 28 °C (*National Data Buoy Center, 2015*) and annual average precipitation of 117 ± 2.35 cm per year (*Western Regional Climate Center, 2019*). In September 2016, 13-months into the experiment, Hurricane Hermine made landfall; however, none of the experimental structures were visibly damaged by the storm. Similarly, we did not observe scouring or sediment build-up around the structures, likely because Corrigan's Reef is located in a relatively protected bay that does not experience particularly high water velocities. Furthermore, the degraded oyster bars where our structures were placed are dominated by shell hash substrates that are more stable than sand or clay substrates characteristic of many other oyster restoration sites.

At each of the five oyster bars, we deployed one replicate of each substrate—a concrete reef and a crab trap—spaced about 4.5 m apart within the intertidal zone, where small clumps of live oysters were observed on the bar, indicating the potential for oyster settlement and growth (US Army Corps of Engineers permit SAJ-2015-01536(NW-JED)). We measured the elevation of each structure using a real-time kinematic GPS (Trimble Geo 7x; Trimble, Sunnyvale, CA, USA). Concrete structures were cast in the Structures Lab at the University of Florida (Gainesville, FL, USA) and consisted of 12 notched, interlocking forms that assembled into a rectangular structure. The modular design facilitated transportation and deployment, as each piece weighed 7–20 kg, and provided stability against moderate wave action. The concrete mix design was standard, with an approximate ratio of 1:2:5:3 for water, cement, course aggregate, and fine aggregate

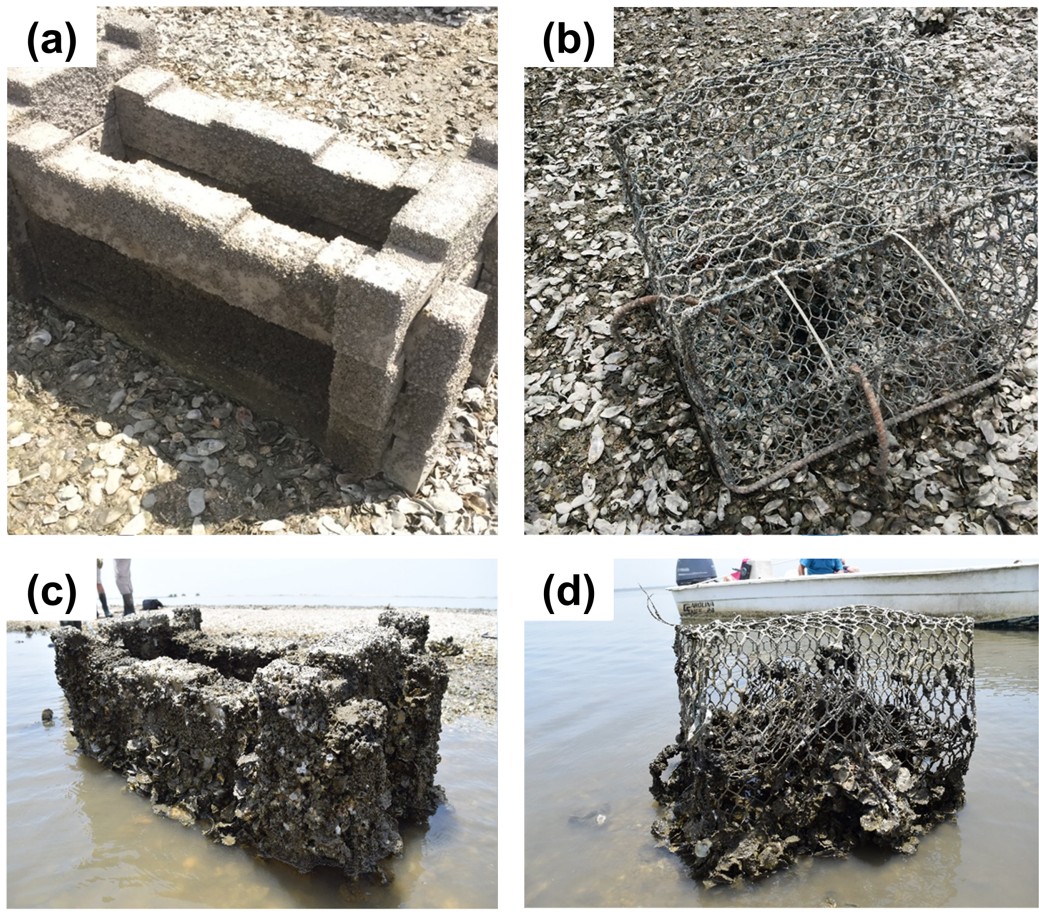

**Figure 1 Photos of concrete and crab trap treatments at the start and end of the experiment.** Interlocking concrete structures (A) and crab traps (B) at the start of the experiment and 17-months later (C, D).

by weight (see Supplement for construction details). When assembled, the structure exterior measured 42 cm H × 96 cm L × 57 cm W (0.23 m³; Fig. 1A).

The plastic-coated wire crab traps were collected from Florida Fish and Wildlife Conservation Commission after their 2015 derelict trap clean-up, cleaned to remove any fouling organisms, and secured to the substrate using two, one meter-long rebar poles whose ends were bent into hooks. Four crab traps were 42 cm H × 61 cm L × 61 cm W (0.16 m³; Fig. 1B), and the fifth was 26 cm H × 120 cm L × 61 cm W (0.19 m³). The one trap with alternative dimensions was used in the experiment at the request of restoration practitioners we work alongside, to evaluate whether this trap type might also perform well in oyster restoration; however, because the mesh size was the same for all traps (five × two cm mesh) and the majority of oysters settled on the bottom 25 cm of the traps, we do not distinguish this trap from the others in our analyses. We closed all crab trap openings using cable ties to prevent traps from capturing terrapins, blue crabs, and other larger species; no dead animals were observed in the traps over the experiment's duration. In addition, we recorded no visible deterioration of the crab traps; however, one concrete structure had one piece knocked askew, likely by a boat, during the experiment.

## Data collection

Data were collected onsite on September 20, 2015 (1-month after deployment), May 24, 2016 (9-months after deployment), August 11, 2016 (12-months after deployment), and June 1, 2017 (22-months after deployment and immediately prior to harvesting). To assess oyster establishment on the structures, we monitored the percent cover of oysters by haphazardly positioning replicate 15 by 15 cm frame quadrats on the seaward- and landward-facing sides of each structure, between 5 and 15 cm from the sediment surface. We used six quadrats on each side (seaward and landward) of each concrete structure, and four and two quadrats on each side of the 0.16 and 0.19 m³ crab traps, respectively. Within each quadrats, we monitored the percent cover by barnacles to evaluate its impact on oyster recruitment and settlement and to compare barnacle establishment rates between the two treatments. Other potentially fouling species known to hinder oyster restoration success, such as sponges and ascidians, were never observed on the restoration structures. In addition, predatory whelks and conchs are common predators of oysters in the region but were not observed on the restoration structures over the course of the experiment so were not monitored.

On the final observation date, we measured the dimensions of the oyster reef established on each structure. Specifically, we took 10 measurements of oyster reef height (i.e., the distance between the benthos and the top of the highest live oyster) and eight measurements of each reef's length and width at heights between 5 and 30 cm above the benthos. We estimated the percent change in the volume of each reef by subtracting the initial structure volume (product of length, width, and height) from the total volume of each reef (product of mean oyster reef length, width, and height measurements) and dividing this difference by the initial structure volume. Structures were then removed from the field, wrapped in tarps to protect the oysters established on them, and transported to University of Florida's Coastal Engineering Lab. In the lab, we used 12 replicate 15 by 15 cm quadrats per structure to measure the percent cover of oysters and barnacles and, within each, counted the number of spat (<2.5 cm), juvenile (2.5–7.5 cm), and market-size oysters (>7.5 cm). We recorded densities of all three sizes classes because they provide unique demographic information about realized oyster recruitment, survival and population development, and sex (i.e., oysters are sequential hermaphrodites that transition from male to female as they get larger, *Thompson et al., 1996*). We also estimated oyster biomass per unit area by removing, oven-drying to a constant weight, and weighing oysters collected from three randomly selected 15 by 15 cm quadrats on each structure.

## Data analysis

Data were analyzed in R 3.4.3 using package "mgcv" (*Wood, 2011*, *2016*). To test for a significant difference in the percent oyster cover between substrate types over time ($\alpha = 0.05$), we specified a set of generalized linear mixed models (GLMMs) that assumed data were beta-distributed (logit link function) and treated the observation date as a random effect (*Breslow & Clayton, 1993*). The full model included fixed effects for substrate type (concrete or crab trap), substrate side (landward or seaward), reef elevation (m NGVD 29), percent barnacle cover, and their interactions. Parameters were estimated
using the restricted maximum likelihood (REML) method. We compared the full model to reduced and null models using Akaike's information criterion (AIC) and report results from the model achieving the lowest AIC value (*Akaike, 1974*). To test for a significant difference in the percent barnacle cover between treatments over time, we used a similar GLMM approach, substituting barnacle cover for oyster cover as the response variable and treating oyster cover as a fixed effect.

The above GLMMs provide beta regression coefficients, which are the odds ratios of the mean response, holding all other variables constant (*Ferrari & Cribari-Neto, 2004*). An odds ratio $P/(1–P)$ gives the probability, $P$, that an event occurs, divided by the probability that the event does not occur. Thus, if an event is three times as likely to occur as to not occur, the probability of the event occurring is $P = 75\%$ (odds = 0.75/0.25). Each beta regression coefficient estimate $\hat{\beta}$ is the logarithm of the odds ratio for a regressor, and from $\hat{\beta}$ we estimate the probability of an event occurring due to a unit increase in the regressor (or, in the case of a categorical variable, due to switching from one factor level to another). That is, from $\beta = \ln\left(\frac{P}{1-P}\right)$ it follows that $p = \frac{e^\beta}{1+e^\beta}$. For instance, if $\hat{\beta} = 2$, the probability that the response occurs increases by an estimated 88% given a unit increase in the regressor.

To test for significant differences ($\alpha = 0.05$) in oyster abundance by size class (spat, juvenile, market-size, and total), reef biomass, and reef volume between substrate types, we specified generalized additive models that assumed the response variables exhibited a Gaussian distribution and used the REML method (*Hastie & Tibshirani, 1990*). The full models for oyster density by size class included structure type, barnacle cover, elevation, and their interactions as fixed factors; full models for oyster biomass and oyster reef volume included structure type, elevation, and their interactions as fixed factors. For each hypothesis test, we report results from the model achieving the lowest AIC value.

## RESULTS

### Oyster cover over time

The GLMM with the lowest AIC value indicated that substrate type, elevation, and the substrate type-barnacle interaction mediated oyster cover during the 22-month experiment (Fig. 2A; Table 1). Importantly, oyster cover was 96% more likely to be higher on concrete structures than on crab traps over time. Oyster cover decreased with increasing elevation but did not differ significantly between the landward and seaward sides of the structures. On concrete structures, barnacle cover had a significant negative effect on oyster cover. The observation date (random effect) was significant, reflecting an increase in oyster cover on both substrate types over the experiment's duration (effective degrees of freedom = 1.94, Chi-square = 70.9, $p < 0.001$, Fig. 1).

After detecting significant positive collinearity between the two numerical predictors in the model—elevation and barnacle cover ($r = 0.278$, $p < 0.001$)—we checked the robustness of the above results against those from a model that excluded one of these predictors. Since oyster cover was more highly correlated with barnacle cover ($r = −0.654$, $p < 0.001$) than with elevation ($r = −0.259$, $p < 0.001$), we excluded elevation as a fixed

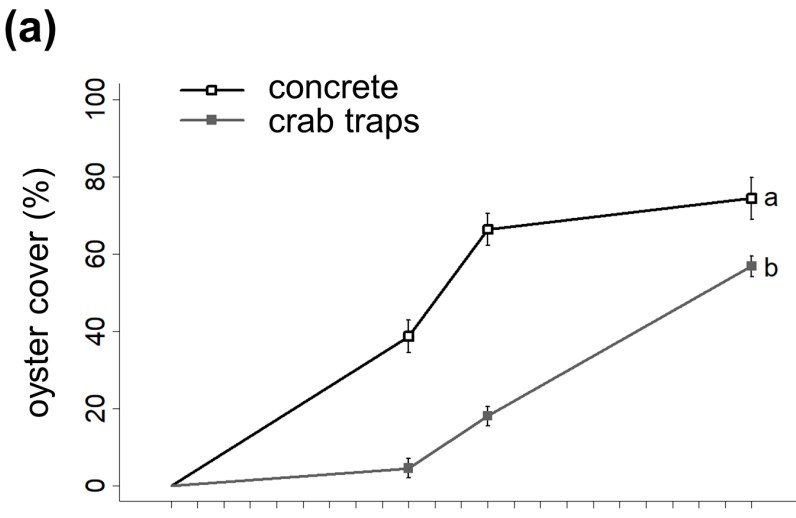

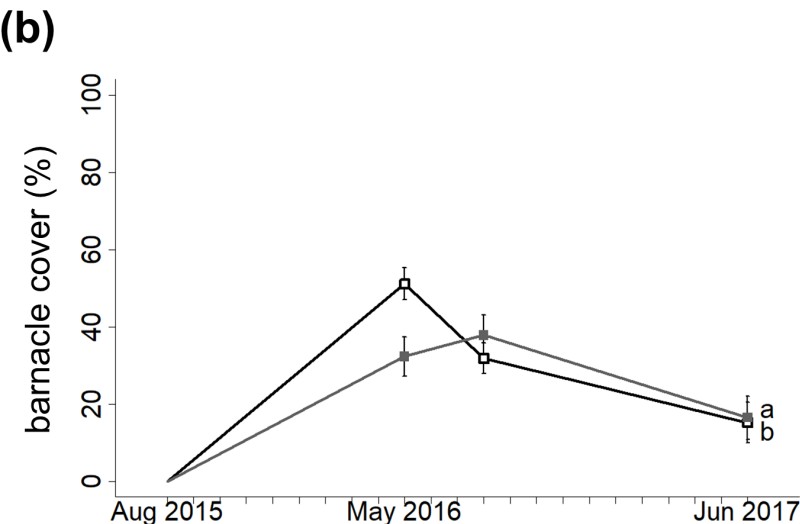

**Figure 2 Changes in oyster cover and barnacle cover over time.** Changes in oyster percent cover (A) and in barnacle percent cover (B) on concrete (open circles) and crab traps (solid squares) over the course of the experiment. Within each panel, different letters indicate a significant difference ($\alpha = 0.05$).

effect in the alternative model. The results from this alternative model were consistent with those reported in Table 1.

## Barnacle cover over time

The GLMM with the lowest AIC value indicated that substrate type, oyster cover, elevation, and the substrate type-oyster cover interaction mediated barnacle cover over time (Fig. 2B; Table 1). Barnacle cover was 87% more likely to be higher on concrete structures than on crab traps over time, and barnacle cover increased with elevation. As with oyster cover, barnacle cover did not differ significantly between the landward and seaward sides of the structures. On concrete structures, oyster cover had a significant

**Table 1  Hypothesis test results for oyster cover and barnacle cover over time.**

| Regressor | Estimate | SE | P | p-value |
|---|---|---|---|---|
| Oyster cover as response | | | | |
| (Intercept) | 2.89 | 1.06 | | 0.007 |
| **Substrate type (concrete)** | **3.23** | **0.21** | **0.962** | **<0.001** |
| Barnacle cover | −0.40 | 0.44 | 0.401 | 0.359 |
| **Elevation** | **−3.90** | **0.94** | **0.020** | **<0.001** |
| **Substrate type (concrete) * barnacle cover** | **−3.26** | **0.48** | **0.037** | **<0.001** |
| Barnacle cover as response | | | | |
| (Intercept) | −3.55 | 0.97 | | <0.001 |
| **Substrate type (concrete)** | **1.91** | **0.21** | **0.871** | **<0.001** |
| **Oyster cover** | **−1.56** | **0.41** | **0.174** | **<0.001** |
| **Elevation** | **3.47** | **0.92** | **0.970** | **<0.001** |
| **Substrate type (concrete) * oyster cover** | **−3.09** | **0.43** | **0.043** | **<0.001** |

Note:
Coefficient estimates are log-odds ratios. *P* is the probability of an increase in the response variable given a unit increase in the regressor. Negative estimates are associated with low probabilities, indicating high probabilities (1−*P*) of negative effects on the response. Significant fixed effects ($\alpha = 0.05$) are indicated in bold.

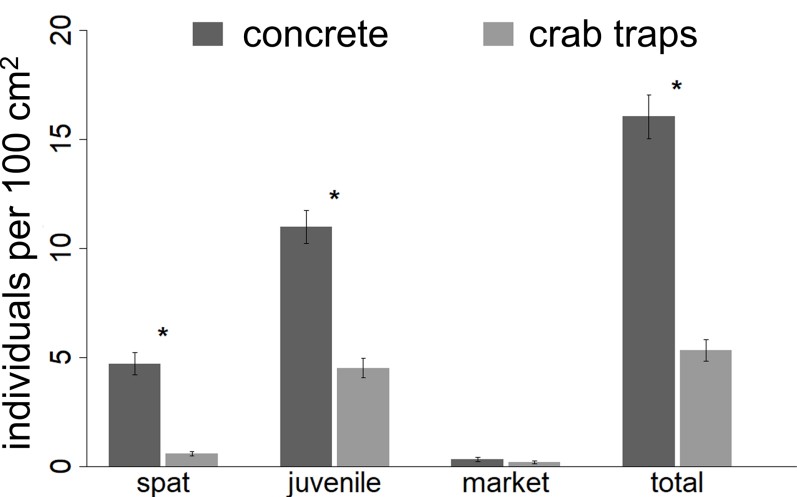

**Figure 3  Oyster density recorded at the end of the experiment.** The number of individual spat, juvenile oysters, market sized oysters and the total number of oysters recorded within 10 by 10 cm square quadrate at the end of the experiment. Asterisks indicate significant differences ($\alpha = 0.05$) between treatments.                               

negative effect on barnacle cover. The observation date was not significant (effective degrees of freedom = 1.11, Chi-square = 2.73, $p = 0.086$).

We detected significant negative collinearity between the two numerical predictors—elevation and oyster cover ($r = -0.259$, $p < 0.001$)—and checked the robustness of the above results against those from a simpler model that excluded elevation as a fixed effect, since barnacle cover was more highly correlated with oyster cover than with elevation. The results from the alternative model were consistent with those reported in Table 1.

**Table 2 Oyster abundance, biomass, and reef volume statistical summary.**

| Regressor | Estimate | SE | p-value |
|---|---|---|---|
| **Spat density (individuals per 100 cm²) as response** | | | |
| (Intercept) | 0.45 | 0.47 | 0.343 |
| **Substrate type (concrete)** | **5.09** | **0.67** | **<0.001** |
| Barnacle cover | 1.28 | 2.47 | 0.606 |
| **Substrate type (concrete) * barnacle cover** | **−6.69** | **3.28** | **0.044** |
| **Juvenile oyster density (individuals per 100 cm²) as response** | | | |
| (Intercept) | 14.92 | 6.60 | 0.026 |
| **Substrate type (concrete)** | **9.72** | **1.09** | **<0.001** |
| Barnacle cover | 4.97 | 3.99 | 0.215 |
| Elevation | −10.74 | 6.36 | 0.094 |
| **Substrate type (concrete) * barnacle cover** | **−19.79** | **5.35** | **<0.001** |
| **Market-size oyster density (individuals per 100 cm²) as response** | | | |
| (Intercept) | 0.82 | 0.90 | 0.361 |
| Substrate type (concrete) | 0.15 | 0.12 | 0.208 |
| Elevation | −0.60 | 0.87 | 0.493 |
| **Total oyster density (individuals per 100 cm²) as response** | | | |
| (Intercept) | 17.27 | 8.25 | 0.038 |
| **Substrate type (concrete)** | **15.11** | **1.37** | **<0.001** |
| Barnacle cover | 6.54 | 4.99 | 0.192 |
| Elevation | −12.44 | 7.94 | 0.120 |
| **Substrate type (concrete) * barnacle cover** | **−27.25** | **6.68** | **<0.001** |
| **Oyster biomass (g per 100 cm²) as response** | | | |
| (Intercept) | 185.47 | 37.42 | <0.001 |
| Substrate type (concrete) | 80.64 | 52.92 | 0.139 |
| **Oyster reef volume (percent change) as response** | | | |
| (Intercept) | 234.78 | 56.53 | 0.004 |
| **Substrate type (concrete)** | **−17.84** | **7.31** | **0.045** |
| **Elevation** | **−183.44** | **54.92** | **0.012** |

Note:
Hypothesis test results for oyster abundance, biomass, and reef volume on the final observation date. Significant factors (α = 0.05) are indicated in bold.

## Oyster abundance by size class

At the experiment's conclusion, spat, juvenile oyster, and total oyster abundances (individuals per 100 cm²) observed on concrete structures were 7.8, 2.4, and 3.0 times higher than those observed on crab traps (Fig. 3; Table 2). Further, barnacle cover had a negative effect on the final abundance of spat, juvenile oysters, and total oysters, but only on concrete structures. The abundance of market-size oysters (i.e., those >7.5 cm) was relatively low (mean ± SE: 0.27 ± 0.06 individuals per 100 cm² across all structures) and did not differ between substrates.

## Oyster reef volume

The percent change in oyster reef volume was significantly higher on crab traps (mean ± SE: 46.7 ± 7.6% increase in volume) than on concrete structures

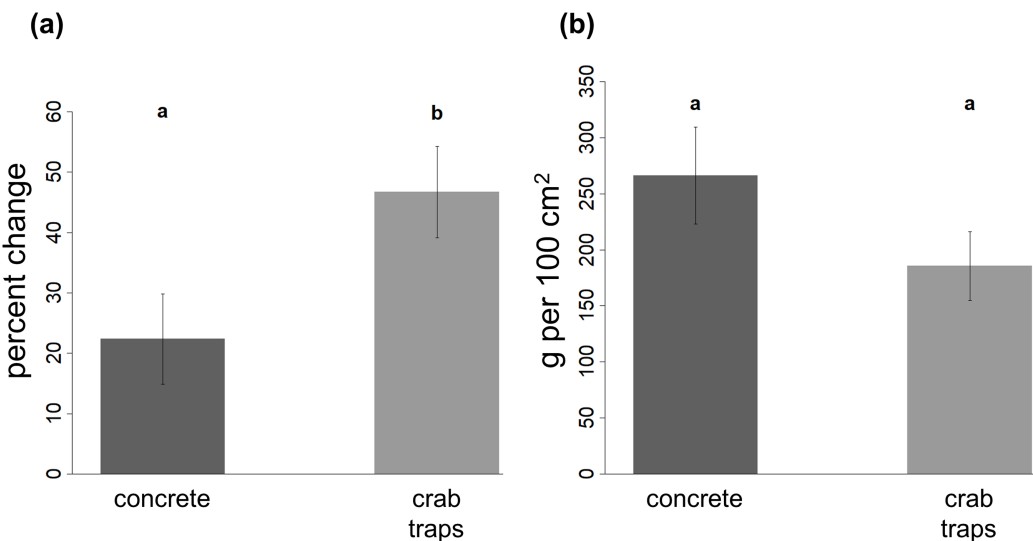

**Figure 4 Change in oyster reef volume and final oyster biomass on restoration structures.** The percent change in oyster reef volume recorded on the concrete and crab trap structures (A) and the final oyster biomass recorded in a 10 by 10 cm quadrat (B) at the end of the experiment. Within each panel, different letters indicate a significant difference ($\alpha$ = 0.05).

(mean ± SE: 22.4 ± 7.5% increase in volume) and decreased with increasing elevation (Fig. 4A; Table 2).

## Oyster reef biomass

Oyster biomass did not vary between substrate types or with elevation (Table 2). Mean (±SE) final biomass was 266.1 (±43.1) g per 100 cm$^2$ on concrete structures and 185.5 (±30.7) g per 100 cm$^2$ on crab traps (Fig. 4B).

## DISCUSSION

Our results reveal that concrete structures and crab traps facilitated oyster reef development over the experiment's almost 2-year period, suggesting that both substrates offer promising solutions for Eastern oyster restoration. In line with our first hypothesis, oysters colonized the concrete structures more quickly than the crab traps (Fig. 2A), and the more rapid establishment resulted in higher densities of oyster spat and juveniles on this material by the experiment's conclusion (Fig. 3). Oysters also established on crab traps and exhibited higher lateral growth on this substrate than on the concrete (Fig. 4A) as predicted by our second hypothesis, indicating that recycled blue crab traps can promote reef formation in micro-tidal, intertidal settings. Barnacles, which can compete with oysters for space, also mediated reef development, such that oysters established relatively quickly and achieved larger reef volumes at lower elevations where barnacles were less abundant and where reefs remained inundated for longer, consistent with our third and fourth hypotheses. Together with our observations that neither structure type deteriorated over the experimental duration (and both survived a hurricane intact), these results suggest that concrete and crab traps are durable, relatively low-cost materials that should be considered for future restoration efforts. In sum,

both experimental substrates successfully jumpstarted reef formation, especially when deployed at elevations conducive to rapid oyster settlement and growth.

In monitoring oyster cover over time, we discovered that the broad, flat surfaces of the concrete were particularly conducive to oyster settlement relative to the crab traps, whose mesh provided far less surface area for establishment (Figs. 1 and 2). The greater availability of settlement surface enabled the concrete to support high oyster recruitment through the 2015 reproductive season (August through October). It is important to note that these results may differ for concrete riprap or crushed concrete, which would have relatively less exposed surface than our interlocking, erect concrete structures. However, we also observed fairly rapid increases in oyster cover on the crab traps during the second summer, likely because those oysters engineered and expanded the available surface area and emitted chemical cues to conspecifics. As a result, there was a near convergence in oyster cover between treatments by the experiment's end (Fig. 2A) and both materials achieved oyster spat, juvenile and adult densities similar or even higher to those reported for concrete and oyster shell-based reefs of a similar age in Texas (*Graham, Palmer & Pollack, 2017*) and in North Carolina (*Theuerkauf, Burke & Lipcius, 2015*). Importantly, these dynamics suggest that materials with high surface areas and ample availability of shaded surfaces—like our interlocking concrete block structures or widely-used oyster shell bags—are not necessarily a prerequisite for successful oyster restoration. Instead, mesh-like materials (e.g., crab traps)—which are typically lighter in weight and thus cheaper to construct, transport, and install—can be viable restoration substrates as long as they support some initial oyster settlement. In addition to their low cost, crab traps are durable and can last 3–7 years in intertidal environments, depending on whether the wire is bare or vinyl-coated (B. Stone & N. Hadley, 2016, personal communication). Furthermore, a South Carolina study found that dipping crab traps in concrete resulted in higher recruitment rates compared to unaltered crab traps, possibly because the concrete creates a textured surface that increases surface area and because calcium carbonate in the concrete may mimic oysters' natural chemical cues (*Kreutzer, 2014*). Before their widespread use in restoration is recommended, additional studies are needed to evaluate the relative performance of bare, vinyl-, and concrete-coated crab traps in different hydrodynamic, oyster larval delivery, and spatial (e.g., intertidal vs. subtidal) contexts.

It is important to note that we only present coverage data from the structures' exterior surfaces, due to the difficulty in quantifying interior oyster and barnacle cover without destroying the structures and the organisms settled upon them. However, we observed significant oyster colonization on the mesh that bisected the interior of the crab traps as well as on the broad interior surfaces of the concrete structures. Inclusion of data from these interior surfaces would certainly alter the inferences drawn from this work to some extent; however, we anticipate that the main factors influencing oyster and barnacle cover—substrate type and elevation—would not change with these interior data. That is, the trends reflected in Fig. 2 are consistent with our observations of the structure interiors: We observed rapid oyster colonization of the interior surfaces of the concrete and more gradual, but substantial, oyster colonization of the

interior mesh of the crab traps (C. Angelini, A. N. Batzer & G. A. Kusel, 2017, personal observations).

By the experiment's end, we observed higher abundances of oyster spat and juveniles, but not market-size oysters, on concrete structures relative to crab traps (Fig. 3). We attribute this difference to earlier and quicker colonization of concrete structures, which likely resulted from the concrete's greater surface area, conspecific feedbacks (i.e., additional surface area and positive chemical cues provided by oysters themselves), and thermal properties (e.g., concrete may keep oysters cooler or increase their vulnerability to heat depending on the duration the structures are inundated by the tides). In general, the sustained, high levels of oyster settlement and growth on both structures indicate that oyster larval loads were quite high during the experiment and are consistent with long-standing knowledge that oysters settle gregariously on conspecifics (*Crisp, 1967*; *O'Beirn et al., 2000*). These results also suggest that there are positive, density-dependent feedbacks whereby strong recruitment and growth in one year can promote and sustain higher oyster densities in subsequent years, as has been documented in longer-term datasets (*Powell et al., 2009*) and proposed in a recent modeling study (*Moore, Puckett & Schreiber, 2018*).

Interestingly, despite differences in oyster cover and in spat and juvenile densities, market-size oysters were relatively rare, with similar densities between substrates. This finding coincides with prior work documenting that oysters in this region typically reach market size within two years (*Wilber, 1992*) and that density-dependent growth and/or predation by oyster toadfish, blue crabs, oyster drills, and other common oyster predators on the concrete structures may have prevented more individuals from reaching market size and transitioning to females. Furthermore, pathogens that have particularly strong effects on larger individuals and during drier years, such as *Perkinsus marinus* (*La Peyre et al., 2003*, *Soniat et al., 2006*), may have contributed to mortality in market-size oysters. We anticipate that a longer experiment would have shown a larger proportion of market-size females than observed.

Additionally, we observed that, despite fewer individual oysters per unit area, the final reef volume was greater on crab traps than on concrete, while oyster biomass per unit area was the same (Fig. 4). These results reveal that the oyster reefs proliferating on each structure type differed in their morphological structure: The concrete-based reefs were shallow and compact, while the reefs that formed on the crab traps were sparser but deeper. We suspect these differences in reef morphology arose due their differential surface areas and effects on water flow. Specifically, while the larger surface area of the concrete enabled these structures to become blanketed in oyster spat during the first year, subsequent growth may have been reduced by the propensity of the wide, flat concrete faces to direct flows around, rather than through, the structure (dynamics that would differ for concrete riprap or crushed concrete). This, in turn, may have hampered delivery of food to settled oysters and thus slowed the reef's ability to extend laterally (*Malouf & Breese, 1977*; *Grizzle, Langan & Howell, 1992*). In contrast, although fewer oysters could settle on the mesh of the crab traps, those individuals that were able to establish may have grown more

quickly due to less restricted water flow and food delivery, thus creating reefs with higher vertical relief and higher-quality predation refuges for spat (*Soniat, Finelli & Ruiz, 2004*). In addition to these physical factors, it is possible that oysters experienced different levels and types of predation on the two substrates such as that imposed by oyster drills and blue crabs, and thus top-down control may have contributed to driving the morphological differences in reef structure. Over longer timescales and in subtidal environments where crab traps have—to our knowledge—never been utilized for restoration, these differences in reef structure may influence the relative "success" of each substrate in promoting the growth and expansion of living oyster reefs beyond the initial footprint of the substrates deployed.

Finally, we observed greater oyster cover, oyster densities, and oyster reef volumes at lower elevations relative to mean sea level (Tables 1 and 2). This finding coincides with results from several other oyster restoration studies (*Lenihan, Peterson & Allen, 1998*; *Bartol, Mann & Luckenbach, 1999*) and natural reefs (*Powell et al., 2009*) documenting pronounced variation in oyster settlement, mortality, and growth rates with elevation. Consistent with conclusions drawn from this earlier work, our results highlight the value of investing in preliminary studies to identify optimal elevational zones prior to deployment of restoration material. Such baseline data, which is needed for both intertidal and subtidal oyster restoration, will likely be essential to ensuring project success and positive returns on investment with regards to oyster growth and ecosystem service provisioning (*Bersoza Hernández et al., 2018*).

## CONCLUSIONS

Our results contribute to an expanding body of work in environmental engineering and sustainable materials design exploring how waste materials from one sector can be repurposed and recycled for environmental benefit (*Fiskel, 2006*). Although our concrete structures were newly cast for this project, concrete is a common waste product with strong potential for oyster reef restoration, as long as the concrete is adequately cleaned and decontaminated. Similarly, abandoned crab traps—which can obstruct boat traffic and continue to trap and kill organisms—can provide a valuable substrate to facilitate the establishment of new living reefs (*Kreutzer, 2014*). Repurposing these waste materials as restoration substrates can reduce burdens on waste facilities, reduce costs associated with restoration, and ultimately restore ecosystem services provided by oyster reefs, including carbon sequestration, water quality enhancement, and habitat for ecologically and commercially important faunal communities (*Grabowski et al., 2012*). More generally, our work demonstrates the viability of alternatives to oyster shell substrates for jumpstarting and sustaining oyster recovery in the Gulf of Mexico.

## ACKNOWLEDGEMENTS

The authors thank Chris Ferraro, Enrique Garcia, and Taylor Humbarger at the University of Florida Structures Lab for their help in constructing the molds and

casting the concrete structures. We also thank Jerry and Laura Adams for sharing their expertise and shuttling us to Corrigan's Reef to collect data.

### Funding

This work was supported by the National Science Foundation CBET CAREER award (No. 1652628) to Christine Angelini. The funders had no role in study design, data collection and analysis, decision to publish, or preparation of the manuscript.

### Grant Disclosure

The following grant information was disclosed by the authors:
National Science Foundation CBET CAREER award: 1652628.

### Competing Interests

The authors declare that they have no competing interests.

### Author Contributions

- Emma E. Johnson conceived and designed the experiments, performed the experiments, prepared figures and/or tables, authored or reviewed drafts of the paper, approved the final draft.
- Miles D. Medina conceived and designed the experiments, performed the experiments, analyzed the data, prepared figures and/or tables, authored or reviewed drafts of the paper, approved the final draft.
- Ada C. Bersoza Hernandez performed the experiments, authored or reviewed drafts of the paper, approved the final draft.
- Gregory A. Kusel performed the experiments, approved the final draft.
- Audrey N. Batzer performed the experiments, approved the final draft.
- Christine Angelini conceived and designed the experiments, performed the experiments, prepared figures and/or tables, authored or reviewed drafts of the paper, approved the final draft.

### Field Study Permissions

The following information was supplied relating to field study approvals (i.e., approving body and any reference numbers):

The field experiment was approved by the US Army Corps of Engineers under permit number SAJ-2015-01536(NW-JED).

### Data Availability

The raw data is available in the Supplemental Files.

### Supplemental Information

Supplemental information for this article can be found online at http://dx.doi.org/10.7717/peerj.6488#supplemental-information.

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
