# Peer review of "Success of concrete and crab traps in facilitating Eastern oyster recruitment and reef development"

_PeerJ, doi:10.7717/peerj.6488_

## Round 0.1 · original submission · Major Revisions

Setting aside the moral or ethical issue of whether waste materials should be deployed in oyster restoration, the major scientific outcome of this study is an assessment of two substrate types for oyster reef development.

All three reviewers make some very valid points around the broader applicability of this study to other regions, the absence of oyster shell as a benchmark comparison, potential use of a broader suite of metrics to assess effectiveness, and founding the study in the context of important ecological processes.

I encourage you to respond to the constructive comments provided, and submit a revised review of your manuscript.

Reviewer 1 ·

Basic reporting

This is an interesting manuscript investigating the use of two different substrates for oyster reef restoration. The results from this study are highly applicable to scientists and restoration practitioners in the Nature Coast. Although the Introduction provides context for the research question, there are quite a few typing errors, missing references, use of extra words and lack of flow between topics and paragraphs that could easily be fixed with a few more rounds of rigorous editing by the authors. The Figures and Tables are relevant and easy to understand but see specific comments and suggestions below.

Experimental design

This article presents original primary research and the methods are easy to understand. The methods are provided with sufficient detail and information to replicate. See additional comments below.

Validity of the findings

The Discussion of this paper is quite strong and results are linked to previous research. The impact of these findings could be improved with further explanation of key research areas that are the foundation to this type of research (i.e. long history of testing oyster reef restoration substrate materials). Specific comments included below.

Additional comments

Major comments:
There is a large body of literature investigating oyster recruitment and reef development in response to a variety of substrates. There is some discussion of substrates in the second paragraph of the Introduction (lines 61-75) but this is mainly to do with reef restoration over longer time scales when in this study oyster recruitment and growth were monitored over a 22 month period. The authors should provide further background on the mechanisms (i.e. shading, chemical cues) that might influence initial oyster recruitment to concrete or wire substrates as these initial differences will have a large impact on overall community development.
Similarly, further justification for the decision not to control substrate material (by dipping crab traps in concrete) is needed. There is reference to a few research chapters that have done this (though one reference is missing, see below) but without further description of the results from these other studies, the results from this study seem premature. For example, did the other researchers find similar recruitment to and growth on concrete coated crab traps vs concrete blocks or concrete coated crab traps vs bare wire crab traps?
Generally, there are a few key concepts in oyster reef restoration, oyster settlement and growth that should be further described. They are: the impact of substrate type on recruitment and reef development; how chemical cues of other substrates (i.e. non oyster material) influences oyster settlement; oyster recruitment on shaded vs open surfaces; and how the vertical height of reefs and restoration material may influence oyster feeding, growth or susceptibility to disease. Additionally, statements about the longevity of using bare wire crab traps rather than concrete dipped crab traps would help make the results of this manuscript applicable to restoration practitioners domestically and internationally. The authors mention that the wire crab traps survived a hurricane, but realistically how long do these wire traps stay intact inter or subtidally? Derelict crab trap clean up programs might be able to help estimate this duration. Where solid concrete, or rip rap based structures are generally good foundations for oyster reef restoration, information about the durability of crab traps as the foundation of reefs will allow scientists and practitioners to determine if this in fact a suitable alternative.
Some references that might be helpful are: Dunn et al. 2014 “Effect of substrate type on demographic rates of Eastern oyster (Crassostrea virginica)”; Soniat et al. 2004 “Vertical structure and predator refuge mediate oyster reef development and community dynamics”
Line 31: Missing “the” in “However, due to relatively open structure”
Lines 56-59: This sentence should be restructured or split in two to improve flow between the previous sentence and the next paragraph.
Line 60: Try to not to start sentences, let alone paragraphs with “Because”
Lines 62-63: Check for consistency in spelling of referenced author “O’Beirn”
Line 64: Are there references to support this sentence? References from the previous sentence likely mention these material so the two sentences could be restructured.
Lines 106: change “low costs” to “low cost”
Lines 108-110: Further explanation of this concept needed. How does dipping crab traps in concrete enhance their surface area? Does it strengthen it? Provide chemical cues more similar to oyster shell? Improve the longevity of the structure? Etc.
Line 110: Kingsley-Smith 2012 is a very critical citation that I tried to find but it is missing from reference list
Line 115: Based on the introduction it appears that evaluating the use of crab traps in oyster restoration is an unexplored field. Why limit the applicability/findings to the Nature Coast?
Line 127: If I am interpreting this hypothesis right, “intraspecific” should be changed to “interspecific”
Line 130-131: This statement should not be part of the hypothesis and could instead by included in the abstract or conclusions.
Line 143: The deployment date and experiment duration is hidden in this sentence but should be more clearly stated, especially if these results will be used to inform restoration projects. I suggest re-working this first paragraph of the Methods section to state “We deployed this field experiment in August 2015 on five offshore... for a period of 22 months…” to improve clarity. Also, I would recommend stating the number of months into the experiment the region was impacted by Hurricane Hermine rather than “mid-way”.
Lines 146-168: This paragraph could be split into multiple paragraphs.
Line 170: Rather than stating just the actual dates monitoring was conducted, the number of months after deployment would also be useful.
Lines 248-250: I find this sentence a bit confusing. Aren’t you more interested in the oyster percent cover? Why this focus on barnacle cover? I agree that barnacle cover is likely to increase if oysters aren’t present but this study was investigating the use of structures to facilitate oyster growth & reef development.
Lines 265-270: I would prefer to read these results within the text rather than a Figure. The percent change in oyster reef volume and final oyster biomass could be presented as in-text values.
Line 308: There is reference to “main factors influencing oyster and barnacle cover” that were identified elsewhere in the Results or Discussion. What are these?
Lines 324-333: There is a large focus on the production of market-sized oysters in this section. However, is the deployment of these restoration structures (concrete or crab trap) intended to provide harvestable oyster populations or instead provide healthy oyster populations that can supply larvae to nearby harvesting reefs? If these restored reefs aren’t intended to be harvested, then we should instead be interested in when oysters reach sexual maturity (i.e. in the juvenile oyster size range) rather than market size. If these reefs are intended for harvesting, then this should be clearly explained in the introduction. This would also lead to further questions I have about the durability and longevity of using wire crab traps if they must also withstand hand-harvesting pressures.
Line 333: remove “however” at the end of the sentence.
Line 360: This sentence starts to address some of the questions I have about the differences in vertical height between the concrete and crab traps and how this influences reef development and oyster survival, but is not fully explained. Rather than just referencing other papers that have discussed how elevation or vertical reef height influences oyster settlement, mortality and growth rates, can you also provide some explanation as to why? Is it because of increased temperature or desiccation stress? Altered water flow or wave exposure? This could perhaps be discussed in a separate paragraph within the Discusison.
Line 491: La Peyre et al. 2014 reference out of place.

Reviewer 2 ·

Basic reporting

I congratulate the authors on a well constructed manuscript, which I found pleasant to read and easy to interpret. The language was unambiguous, but at times a little long-winded. More concise, active voice would make parts of the introduction more engaging.

Comments by line:
Line 55 – as written it is slightly ambiguous what Grabowski’s evaluation relates to here. I would say something like “services which are collectively valued between…”

Line 56-57 – wow that is impressive!

Line 60-61 – true, hard substrate availability is the major limiting factor to recovery where there remains high spat fall (such as the southern states), but at many sites water quality, disease and limited spat fall restrict recovery. Suggest you clarify you are talking about a system where spat is not the limiting factor, at which sites provision of hard substrate is the primary strategy for restoration.

Paragraph starting line 76 – this paragraph is very region specific and of limited interest to the broader readership. I think generally discussing the environmental change/threats that are relevant to oyster decline in this area (i.e. changing freshwater flows, sea level rise) may appeal to more readers. You can then discuss the local context later in the introduction, and how this region is an area where these threats must be mitigated.

Line 106 – low “cost”, no plural

Line 106-108 – more concise language could reduce this sentence by about 10 words. E.g. “Repurposed crab traps are cost-effective structures for oyster restoration because their expense is limited to transportation and installation costs”.

Line 115 – “never been valued in the Nature Coast”. Again, these paragraphs are very site specific, which will limit the ever growing international readership interested in oyster restoration.

Line 120 – I am glad you monitored intraspecific recruitment of barnacles, but can you extend this to other fauna? One of the primary concerns of deploying substrate/structure in the environment is whether it unintentionally facilitates “undesirable” species, namely invasives. Surveying the complete faunal assemblage recruit to each structure could address this. Furthermore, from a restoration viewpoint, where we are trying to enhance the fitness of the oysters, I would be particularly interested in whether these structures facilitated oyster predators, such as crabs, which could negatively impact oyster performance.

Line 144 – that is a positive result if the concrete structures can withstand hurricane forces. Did the hurricane not impact the crab traps? As hurricanes are a persistent issue in the region and you inadvertently examined their impact on restoration structures, do you discuss this further?

Line 157 – “likely similar to other concrete-based materials”…a point of reference for what you expect will be similar to other concrete structures. Oyster abundance/m2?

Line 333 – remove “however”

349-341 – I am glad you mentioned this. To my mind this is a missed opportunity for your study, because quantifying predation (structures attracting predators vs provision of a predation refuge – inside the crab trap looks like a good place to live!) would allow restoration practitioners in different systems to apply this knowledge to their known compliment of predators.

368 – I think it’s tricky to talk about “sustainable materials” – perhaps true for abandoned crab traps, certainly not for concrete production. I think “opportunistic materials” is more appropriate.

Line 371-372 – “…recycled for oyster restoration as long.” As long what? Advocating hauling concrete from demolished buildings to place in the intertidal for restoration sounds like a hard sell. There may be opportunities from time to time, but not as a environmental management strategy.

Line 374: “obstuct” . Also, couldn’t they potentially trap fauna once they partially erode?

Figures could be sharpened up a little by greater distinction between colours on the bar graphs, and fewer tick marks on the x-axis of the Fig. 2. In the legend for Fig. 2, should it say “5 treatment replicates”, as it is the substrate treatment that is replicated. As it reads it suggests there are 5 treatments per substrate type.

Experimental design

I think the design and execution is fine if purely interested in oyster-structure interaction. But I feel not collecting more biological (facilitated species assemblage, predation rates) or environmental data (temperature differences between the intertidal structures, sediment scour/erosion impacts of structures) is a missed opportunity to expand the relevance of this study to other habitats and biological systems.

Line 163-164 - A short explanation as to why all crab traps were not the same size would be good.

Line 170 – I’m a little confused about the sampling frequency and dates. On line 143-144 you state the experiment ran from Aug 2015 till May 2017, though you sampled after this in June 2017 (line 171). I am struggling to see a defined pattern for your sampling frequency, so a little clarity on this would be good. Personally, I find specific dates like this distracting. I would be more inclined to say something like “After establishing the experiment in August 2015, we initially collected data after one month, and then approximately every 12 months until it’s completion in …”

Line 175 – that is 6 quadrats were replicate concrete structure? Please clarify

Validity of the findings

The results section and data discussion are clearly written. I am satisfied with the data and information provided in the supplemental materials. The statistical approach is well reasoned.

Conclusions are clearly stated. I particularly like how they hypotheses were partitioned first in the introduction, which help me interpret the results later.

Additional comments

I have two major concerns with this work. The first is the extremely narrow focus on the circumstances of the Nature Coast. Yes the local circumstances drove the design and opportunistic use of crab traps, but the introduction and overall focus needn't be so site specific, which disengages those unfamiliar with your system. I suggest that the discussion of using alternative materials to those typically used to restore oysters (i.e. shell) takes center stage, and your site and region are described as the vehicle to test these materials.

The second issue, and the more limiting, is the missed opportunities to quantify other biological and/or environmental responses to the structures. I believe this is important because environmental restoration efforts should be underpinned by evidence based reasoning. The authors pay little mention to how each structure influenced the surrounding abiotic environment, some impacts of which may be negative. For example, wave action around the intertidal concrete structures would likely scour the surrounding sediment, potentially altering the immediate benthic environment or impacting downstream sites. Further, little attention is paid as to how the two structures provide extremely different environments for the oysters. For example the concrete would likely store substantial heat when exposed to the sub-tropical sun, such that substrate heat stress would impact the oysters. Meanwhile, the crab trap would likely provide a predation refuge from larger predators (i.e. fish and crabs), but it may also facilitate meso-predators (who may also benefit from the predation refuge of the trap. My overarching concern here is, if we advocate the re-purposing of disused concrete (i.e. from demolition sites: line 371) to place in the marine environment, we want to be very sure there will be no negative environmental impacts (e.g. navigation hazard, chemical leaching, facilitating invasive species), otherwise it is more environmental vandalism than restoration.

Reviewer 3 ·

Basic reporting

Overall the manuscript is well written, but in some places the English is a little awkward and could do with some improvement - e.g., lines 42 to 48 - over dramatic language and clunky description of the issues involved - could be streamlined.

Referencing is appropriate and breadth of literature well used

Article structure is sound and self contained.

Experimental design

Fits within aims and scope of journal

Research question is well defined in terms of a knowledge gap, however, it is very unfortunate the experiment did not include a comparison with traditional oyster restoration materials, such as bags of shell so that an appropriate comparison could be made to provide broader relevance.

In terms of the experimental design - the 5 replicate structures for all treatments were all in a relatively confined area so it is hard to assess whether the results have wider applicability other than for this localised site.

One crab cage was a different size and it is not clear how this possible influence on the results was dealt with in the experimental design.

The duration of the experiment is relatively short so the persistence of the structures would be interesting to follow especially at the crab pots were likely to be breaking down.

Ethically I have concerns about altering the natural coastal environment with concrete and waste crab traps- are these zinc coated wire traps which would leave a toxic residue in the marine environment, or plastic coated, which would leave plastic waste in the marine environment. This requires some commentary at least, as in my part of the world dumping concrete and waste crab traps in the coast environs would be unacceptable and would not be permitted by regulatory authorities.

Validity of the findings

The results have value for oyster restoration, but are somewhat limited in scope in my view by the lack of measurement of the results over a wider spatial scale to confirm they are consistent, and not a localized phenomenon. Furthermore, the lack of comparison with oyster shell limits any assessment of the comparative performance with traditional enhancement approaches.

The data is well collected and the analyses look entirely appropriate to me. The results would benefit from reporting of more information on the scale of measured comparative differences in results. e.g., "Similar to oyster cover, barnacle cover also did not vary
between landward and seaward sides of the structures." - we don't actually know what the amount of measured cover is for each of the structures at different heights.

Conclusions are sound and well founded on the data presented. The conclusion I found quite long given the extent of the research results and its relative significance.

Additional comments

This is a well presented manuscript that is of some relevance to oyster restoration efforts. I have outlined some of my points in preceding sections of the review and these should be shared with the authors.

---

## Round 0.2 · Minor Revisions

Thank you for your response to the constructive comments made by three reviewers.

Overall, the manuscript is improved in clarity, however the limits of the experimental design remain – i.e., no experimental comparison of the alternative substrates to oyster shell, and lack of biological or environmental data to determine the mechanisms driving differences in oyster recruitment and reef growth.

While I appreciate that there is no means to go back and redo, the manuscript could still be strengthened. For example, to evaluate this study in light of oyster substrates, a comparison of oyster densities on concrete structures and crab traps versus oyster shell substrates should be included in the discussion. Furthermore, a comment is warranted regarding the regularity of the deployed concrete structures in this study versus the potential irregularity in blocks of demolished concrete.

Another area of the manuscript that still needs improvement is the hypotheses and conclusions (that do not go back and address the hypotheses). In hypothesis 2 for example, given that water flow nor food concentration was measured in the experiment, a null hypothesis may be a better approach, or use of the words ‘could’ or ‘potential’ rather than “due to’ is advised.

I have made a few minor edits to the text (see the attached pDF), and would encourage the authors to resubmit a revised version, having made the suggested amendments.

Minor points:

Use of hectares not SI units (km2) in Introduction – providing both units would be useful
Figure 2: Symbols should be larger and substrates easier to compare (e.g., if the grey line was black dashed this would be easier to tell than black versus dark grey). Also, what do the a and b next to the lines mean? Not explained in legend. Ditto for Figure 4.

---

## Round 0.3 · accepted · Accept

Thank you Christine, for responding to my final comments on your submission. All the best!

#